# Enhanced recombinant protein production in CHO cell continuous cultures under growth-inhibiting conditions is associated with an arrested cell cycle in G1/G0 phase

Verónica Avello[1,2¤], Mauro Torres[3,4], Mauricio Vergara[1], Julio Berrios[1], Norma A. Valdez-Cruz[5], Cristian Acevedo[2,6,7], Maria Molina Sampayo[8], Alan J. Dickson[3,4], Claudia Altamirano[1,9]*

1 Escuela de Ingeniería Bioquímica, Pontificia Universidad Católica de Valparaíso, Valparaíso, Chile, 2 Centro de Biotecnología, Universidad Técnica Federico Santa María, Valparaíso, Chile, 3 Manchester Institute of Biotechnology, Faculty of Science and Engineering, University of Manchester, Manchester, United Kingdom, 4 Department of Chemical Engineering, Biochemical and Bioprocess Engineering Group, University of Manchester, Manchester, United Kingdom, 5 Departamento de Biología Molecular y Biotecnología, Instituto de Investigaciones Biomédicas, Universidad Nacional Autónoma de México, Ciudad de México, México, 6 Centro Científico Tecnológico de Valparaíso CCTVaL, Universidad Técnica Federico Santa María, Valparaíso, Chile, 7 Departamento de Física, Universidad Técnica Federico Santa María, Valparaíso, Chile, 8 Centro de InmunoBiotecnología, Universidad de Chile, Santiago, Chile, 9 Centro Regional de Estudio en Alimentos Saludables, R17A10001, Valparaíso, Chile

¤ Current address: Biotechnology and Biopharmaceutical Laboratory, Pathophysiology Department, School of Biological Sciences, Universidad de Concepción, Concepción, Chile
* claudia.altamirano@pucv.cl

**Data Availability Statement:** All relevant data are within the paper and its Supporting information files.

## Abstract

Low temperature and sodium butyrate (NaBu) are two of the most used productivity-enhancing strategies in CHO cell cultures during biopharmaceutical manufacturing. While these two approaches alter the balance in the reciprocal relationship between cell growth and productivity, we do not fully understand their mechanisms of action beyond a gross cell growth inhibition. Here, we used continuous culture to evaluate the differential effect of low temperature and NaBu supplementation on CHO cell performance and gene expression profile. We found that an increase in cell-productivity under growth-inhibiting conditions was associated with the arrest of cells in the G1/G0 phase. A transcriptome analysis revealed that the molecular mechanisms by which low temperature and NaBu arrested cell cycle in G1/G0 differed from each other through the deregulation of different cell cycle checkpoints and regulators. The individual transcriptome changes in pattern observed in response to low temperature and NaBu were retained when these two strategies were combined, leading to an additive effect in arresting the cell cycle in G1/G0 phase. The findings presented here offer novel molecular insights about the cell cycle regulation during the CHO cell bioprocessing and its implications for increased recombinant protein production. This data provides a background for engineering productivity-enhanced CHO cell lines for continuous manufacturing.

**Funding:** This work was financially supported by the following funding: Biotechnology and Biological Sciences Research Council, BB/N022041/1, Prof Alan J Dickson Agencia Nacional de Investigación y Desarrollo, 1200962, Prof. Claudia Altamirano Agencia Nacional de Investigación y Desarrollo, 11190488, Dr Mauricio Vergara Agencia Nacional de Investigación y Desarrollo, 180146, Prof Maria Molina Sampayo Agencia Nacional de Investigación y Desarrollo, ACT210068, Prof. Claudia Altamirano.

**Competing interests:** The authors have declared that no competing interests exist.

## Introduction

Improving the productivity phenotype of Chinese hamster ovary (CHO) cells is a major challenge in biopharmaceutical manufacturing. Whilst significant advances in cell line development and bioprocess optimisation have catapulted product yields to above 5 g/L [1, 2], these improvements are often restricted to a relatively narrow subset of recombinant molecules, particularly monoclonal antibodies (mAbs). Meanwhile, a large number of therapeutic proteins remain (very) difficult to produce, with product yields of about mg/L level [3–5]. Limited production yields affect the manufacturability of these biopharmaceuticals and restrict access to these life-changing medicines for patients.

Over the past three decades, industrialists and academics have developed a vast repertoire of strategies (both at genetic and process levels) for increasing CHO cell productivities. One of the most successful approaches has been the control of cell proliferation, in which a decrease in cell proliferation (referred to as the cell-specific growth rate) can result in enhanced cell-productivities [6]. On a genetic level, the control of cell proliferation has been achieved by the overexpression of critical regulatory proteins of the cell cycle, a strategy that has been successful in enhancing CHO cell productivity phenotype [7–9]. However, the profound arrest of the cell cycle and cessation of cell proliferation resulted in restrictive conditions from an industrial manufacturing perspective, where cells need to achieve high cell densities in large-scale volumes.

On a process level, the modification of environmental parameters or media supplements has also led to increased cell-productivities by stopping cell growth at different degrees [10, 11]. In this case, the change in culture conditions is usually performed in a biphasic operation, where initial cultivation under 'standard' physiological conditions is followed by a growth-inhibited phase. Two of the most used process engineering approaches to enhance the CHO cell performance are a decrease in temperature (from 37°C to 33–31°C) and supplementation with sodium butyrate (NaBu, from 0.5 to 3 mM) [1, 6, 10]. A large body of research has shown that these strategies generate a differential expression of genes involved in the cell cycle and protein folding/secretion both at a gene and a protein level [12–19]. Such changes have been interpreted to indicate a tight (inverse) relationship between cell growth and r-protein production at a molecular level. However, as most studies have been performed in (fed)batch cultures, it is difficult to isolate the effect of growth inhibition from the specific/individual effect of these two process interventions. This problem limits our understanding of the underlying causes promoting the increased productivities, a scenario that makes the outcomes of low temperature and/or NaBu treated cultures in CHO cell cultures sometimes difficult to predict, thus requiring a laborious process optimisation.

This study aims to question the relationship between growth inhibition and its productivity-enhancing effects in the context of low temperature and NaBu-treated cultures. To address this question, we used a continuous culture system that enabled cell-specific growth rate to be set at a desired value (by controlling the dilution rate) and to evaluate the consequences of decreasing culture temperature an/or NaBu addition under steady-state conditions. In this study, we set a series of chemostat cultures at two dilution rates (0.018 and 0.010 1/h), two temperatures (37 and 33°C), and two medium conditions (with and without NaBu supplementation). All chemostat cultures were operated at a steady-state, and from each cultivation, cell growth, r-protein production, culture metabolites, cell cycle distribution, and transcriptome profile were evaluated. Given the complexity to interpret the results of the evaluated conditions, a multivariate statistical analysis (MANOVA) and correlation analysis were utilised to unravel the individual and combined contribution of dilution rate, low temperature, and NaBu supplementation in the CHO cell performance.

## Material and methods

### Cell line and cell culture

A human recombinant tPA (hr-tPA) expressing CHO cell line was used in this study [20]. The CHO cell line was grown routinely in HyClone SFM4CHO medium (HyClone) supplemented with 20 mM glucose (Sigma) and 6 mM glutamate (Sigma). Maintenance cultures were seeded at a cell concentration of $2.5 \times 10^5$ cells/ml and subcultured every 48 hours using T-75 flask and 10 mL culture medium. Cells were scaled up to 50 mL cultures using 125 mL spinner flasks (Techne, UK). For low-temperature cultures, cells were adapted by exposure to a progressive decrease in temperature. To achieve this, cells were initially cultivated at 35˚C for three culture passages (every 48 h) and then exposed to 33˚C for an additional three passages before use in experiments. The gradual decrease in culture temperature enabled the acclimatisation process by maintaining high cell viability.

Continuous cultures were performed in 250 mL spinner flasks, specially conditioned for a fresh medium inlet, cell culture medium outlet, and a sterile filter for gas exchange [21]. Cells were cultivated in 150 mL HyClone SFM4CHO medium (supplemented with 20 mM glucose and 6 mM glutamine). Cultures were initially operated in batch-mode for 48 h and then supplied with sterile feed (i.e., fresh growth media) throughout the culture operation. All cultures were maintained at 37˚C (except for low-temperature cultures, which were at 33˚C) with shaking at 80 rpm, 5% $CO_2$-enriched atmosphere, and controlled humidity in a Forma Scientific $CO_2$ Incubator (Thermo Fisher Scientific Inc., USA). To evaluate different culture conditions of temperature (37˚C and 33˚C), dilution rate (0.01 1/h, Low dilution rate [LD]; 0.018 1/h, High dilution rate [HD]) and sodium butyrate (0 mM, Basal medium [BM]; 0,5 mM, NaBu supplemented [NB]), eight experiments (in duplicate) were performed as described Fig 1. Cultures were sampled periodically for viable cell quantification and further analytical measurements. Cell cultures were regarded to have reached a steady-state (SS) when, after at least four residence times, both the number of viable cells and lactate concentration were constant in two consecutive samples. Once steady-state was reached, the cultures were perturbed (step perturbation) using 0.5 mM of sodium butyrate, which concentration was selected previously [22].

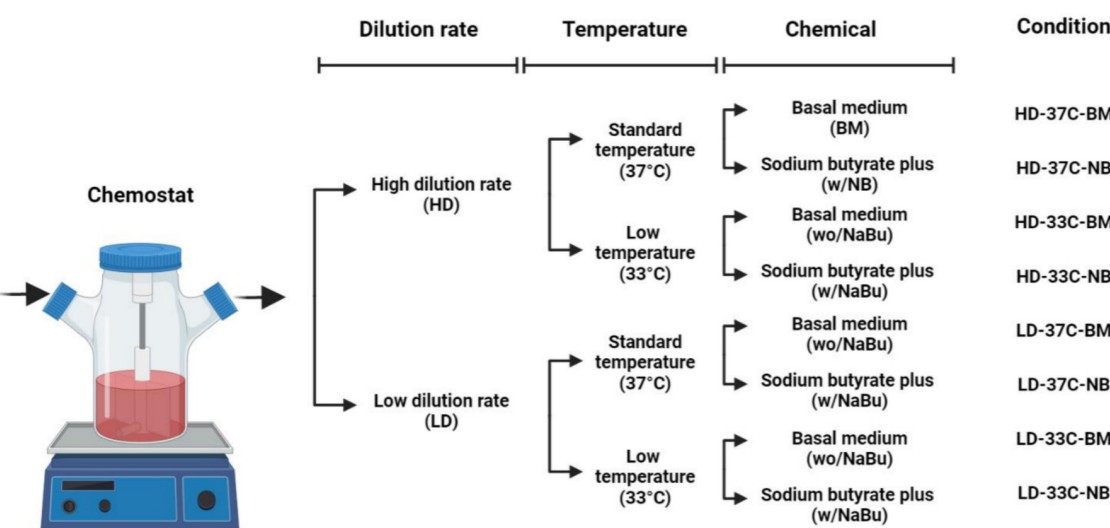

**Fig 1. Schematic representation of the culture conditions performed in continuous cultures.** BM and NB correspond to basal medium with and without sodium butyrate, respectively. HD-37C-BM represents the control culture condition.

## Analytical methods

Cell concentration and viability were determined by hemocytometer analysis (Neubauer, Germany) using the trypan blue exclusion method. Extracellular rh-tPA was quantified by a commercially available enzyme-linked immunosorbent assay (Trinilize tPA-antigen kit, Tcoag Ltd., Ireland). Glucose and lactate were measured with an automatic biochemistry analyzer (YSI 2700, Yellow Springs Inc., USA).

## Cell cycle analysis

For cell cycle analysis, $1.5 \times 10^6$ cells in 200 μL PBS were fixed with 1.8 mL cold 70% (v/v) ethanol, and samples were stored at −20 ˚C until measurement. Fixed cells were mixed with 1 mL propidium iodide solution (40 μg/mL final concentration, Invitrogen, USA) in 3.8 mM sodium citrate (Sigma, USA). After a 3 h incubation at 4 ˚C with 50 μL stock RNase A solution in the dark, cell cycle status was determined using a Beckman Coulter FC500 flow cytometer (Beckman Coulter Life Sciences, USA). Flow cytometry data were analysed using Cylchred software from Cardiff University. Results were presented as the percentage of cells in G1/G0, S, and M phases of the cell cycle.

## Transcriptome analysis

Evaluation of the transcriptomic response of CHO cells was performed using DNA microarrays with RNA samples taken from chemostat cultures during the steady-state. Sample preparation for gene transcripts profiling was performed using GeneChip™ WT PLUS Reagent Kit (Applied Biosystems™), and cDNA synthesis was performed using the GeneChip™ WT PLUS Reagent Kit (Applied Biosystems™) according to the manufacturer's protocols. The sense cDNA was fragmented and biotin-labeled with terminal deoxynucleotidyl transferase (TdT) using the GeneChip™ WT Terminal Labeling Kit (Applied Biosystems™). Approximately 5.5 μg of labeled cDNA target was hybridized to the GeneChip CHO Gene 2.0 ST Array (Affymetrix) at 45˚C for 16 hours. Hybridized arrays were washed and stained on a GeneChip™ Fluidics Station 450 (Applied Biosystems™) and scanned on a GeneChip™ Scanner 3000 7G (Applied Biosystems™). Raw data was automatically extracted using Affymetrix® GeneChip Command Console® Software (Applied Biosystems™). The data was analysed using the Robust Multi-array Average (RMA) method and the software Affymetrix® Expression Console™ Software (Applied Biosystems™). Comparative analysis between the test and control samples was performed using the change times based on the control condition (fold change). Transcripts were associated with known genes using the reference genome of the CHO-K1 cell line [23]. To associate gene expression with molecular pathways, the database of Cricetulus griseus (cge) and Mus musculus (mmu) was used [24]. The raw data on transcriptome analysis is available in S1 File.

## Calculation of specific rates

Cell-specific growth rates were calculated from a mass balance within the bioreactor:

$$\mu = D(N_t/N_v)$$

Where $N_v$ is the concentration of viable cells ($10^6$/mL), $N_t$ is the concentration of total cells ($10^6$/mL), and D is the dilution rate of the culture (1/h) [25].

Specific rates of production or consumption of a metabolite i (qi) were calculated as follows:

$$q_i = D\left(\left(C_i^i - C_i^0\right)/N_v\right) \times 10^9$$

Where $C_i^i$ is the concentration of i in the inlet (mmol/L), $C_i^0$ is the concentration of i in the outlet (mmol/L), $N_v$ is the concentration of viable cells ($10^6$/ml), and D is the dilution rate of the culture (h-1). In the case of NaBu cultures, μ and qi were calculated by least-squares adjustment using the excel solver tool.

## Statistical analysis

All culture experiments were performed in duplicate, and two independent samples were taken at each time point for every culture, with analytical measurements carried out separately. Values of kinetic and stoichiometric parameters are expressed as the mean ± standard error of the mean (SEM). All statistical analyses were performed using R software (version 3.1.). Statistical significance of the variations caused by the different experimental conditions was evaluated by multivariate analysis of variance (MANOVA) (using culture temperature– 2 levels, dilution rate– 2 levels, and NaBu supplementation– 2 levels as factors) followed by multiple comparison tests (Tukey HSD test) with normally distributed data. Variance homogeneity and normal distribution of residuals were assessed with the Shapiro-Wilk test and visual inspection of the normal-quantile plot to validate MANOVA's assumptions.

## Results

### Performance of the hr-tPA producing CHO cell line in continuous culture

In terms of culture performance, we observed the highest viable cell density (VCD) and specific cell growth rate (μ) in the control condition (Fig 2A and 2B). A MANOVA analysis revealed that dilution rate and NaBu had the most significant impact on VCD and μ in CHO cells (Table 1). In particular, we found that LD and NaBu supplementation significantly decreased VCDs compared to the control cultures, a response that was consistent with previous studies in CHO cell cultures [1, 25]. However, low temperature had a different impact on cell growth depending on the dilution rate. Cells showed a decrease in VCD at HD, whereas it had the opposite effect at LD. We also observed that low temperature attenuated the reduction in VCD of LD cultures. This phenomenon seemed to be associated with the improving effects of low temperature cultures on cell viability (S1 File).

In contrast, the control cultures presented the lowest r-protein production and cell-specific productivity (Fig 2C and 2D). We observed that dilution rate, temperature, and NaBu supplementation (both individually and combined) had a significant impact on increasing both hr-tPA titre and $q_P$ in CHO cells (Table 1). The HD-33C-NB and LD-33-NB were the conditions with the highest improvement in product titre and cell-specific productivity of all the conditions. This data showed that low temperature and NaBu supplementation had an additive effect on increasing r-protein production, an observation that has been previously reported in other CHO cell lines [1, 26]. We also found that combining LD and 33˚C presented an additive effect on hr-tPA titre and $q_P$, even though these two approaches only led to slight improvements when applied separately. A correlation analysis showed that hr-tPA production was inversely correlated to cell-specific growth rates in chemostat cultures without NaBu, which agreed with previous studies in CHO cell cultures [27, 28]. However, this correlation was inverted in cultures treated with NaBu (Fig 3). These data suggested that controlling cell growth rates was an important aspect for increasing productivity phenotypes, but there were other aspects beyond the control of cell proliferation that modulate the r-protein production.

We did not observe significant differences in glucose consumption in terms of culture metabolites, but lactate production was strongly affected by the culture conditions. We found that LD decreased the lactate production rate while NaBu supplementation increased it in all

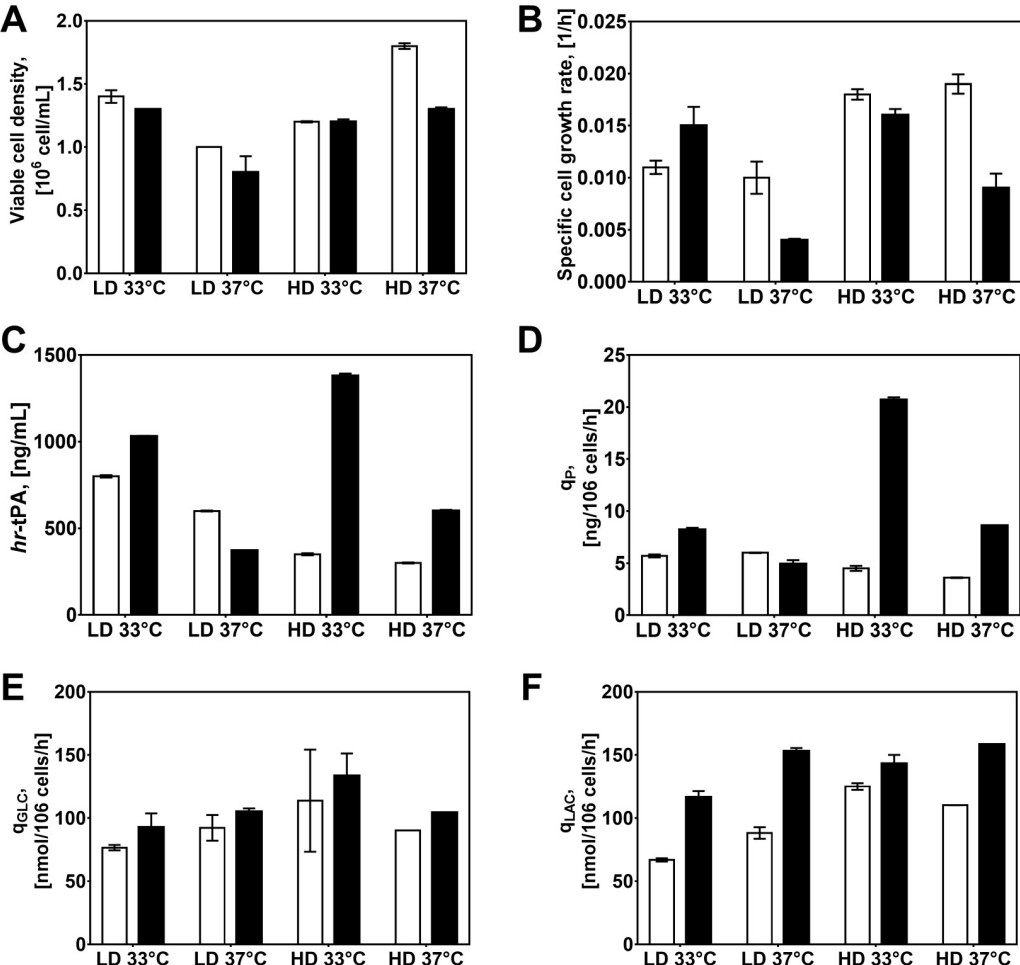

**Fig 2. Culture parameters of CHO cell continuous cultures.** (*white*) without NaBu and (*black*) with NaBu. A) Viable cell density. B) Specific growth rate. C) hr-tPA titre. D) Cell-specific productivity. E) Glucose consumption rate. F) Lactate production rate. Statistical analysis of culture parameters is detailed in Table 1. Experimental values represent the mean of two biological replicates ± SEM. HD, High dilution rate; LD, low dilution rate; GLC, glucose; LAC, lactate; NaBu, sodium butyrate, hr-tPA, human recombinant tissue plasminogen activator. Profiles of CHO cell cultures at steady state are found in S2 Fig in the S2 File.

conditions. We also found that low temperature had opposite effects on cell-specific lactate production ($q_{LAC}$) depending on the dilution rate, with an increase at HD and a decrease in LD. This phenomenon seemed to be linked to the growth rates in each condition, particularly in cultures without NaBu addition. We found that $q_{LAC}$ and μ were strongly correlated in cultures without NaBu (Pearson's correlation, $R_{qLAC}$ *vs* μ = 0.84, p = 0.0087). This finding agreed with previous reports linking lactate production with cell proliferation [29].

## Cell cycle analysis and its relationship with hr-tPA production

Given the differences in growth/productivity relationship in cultures with and without NaBu, we evaluated whether the improvements in hr-tPA titre and cell-specific productivity were associated with changes in cell cycle phases (Fig 4). Cell cycle distribution was analysed by calculating the percentage of cells in G1/G0, S, and G2/M phases for all chemostat cultures at a steady state. In general, we found that each culture condition significantly impacted the cell

**Table 1. Multivariate statistical analysis of the individual and combined impact of dilution rate, culture temperature, and NaBu supplementation on cell culture parameters.**

|  | Dil | Temp | NaBu | Dil:Temp | Dil:NaBu | Temp:NaBu | Dil:Temp:NaBu |
|---|---|---|---|---|---|---|---|
|  | *p*-value | *p*-value | *p*-value | *p*-value | *p*-value | *p*-value | *p*-value |
| VCD | *** | n.s. | ** | *** | n.s. | * | n.s. |
| μ | *** | n.s. | ** | ** | n.s. | n.s. | * |
| $q_{Glc}$ | n.s. | n.s. | n.s. | n.s. | n.s. | n.s. | n.s. |
| $q_{Lac}$ | *** | ** | *** | ** | ** | ** | n.s. |
| Titre | *** | *** | *** | *** | *** | *** | *** |
| $q_P$ | *** | *** | *** | *** | *** | *** | *** |
| G1/G0 | *** | *** | ** | n.s. | n.s. | n.s. | * |
| S | n.s. | *** | *** | n.s. | n.s. | n.s. | ** |
| G2/M | *** | *** | *** | n.s | *** | * | n.s. |

Dil: dilution rate; Temp: culture temperature; NaBu: sodium butyrate supplementation.

***, **, * and n.s correspond to p < 0.001, p < 0.01, p < 0.05 and non-significant, respectively.

cycle distribution of CHO cells (Table 1). For the control cultures, cells presented the lowest and the highest population in G1/G0 and S phases, respectively. A decrease in dilution rate and/or temperature resulted in a significant increase in the number of cells in the G1/G0 phase, with a subsequent decrease in S phase (Fig 4A and 4B). Although the NaBu supplementation also increased the cell population in the G1/G0 phase, we observed an increase in cells in the G2/M phase, particularly at HD (Fig 4A and 4C). These findings agreed with previous studies showing cell cycle arrest in G1/G0 phase in cultures at low temperature or treated with NaBu [1]. We also found that the LD-33C-NB cultures presented the highest amount of cells in G1/G0 phase, suggesting that the approaches to control cell growth can synergise for arresting the cell cycle.

To evaluate the relationship between cell cycle and hr-tPA production, we performed statistical Pearson's correlations between the cell cycle data and the data corresponding to hr-tPA titres and cell-specific productivities (Fig 5). We found that an increase in hr-tPA titre and

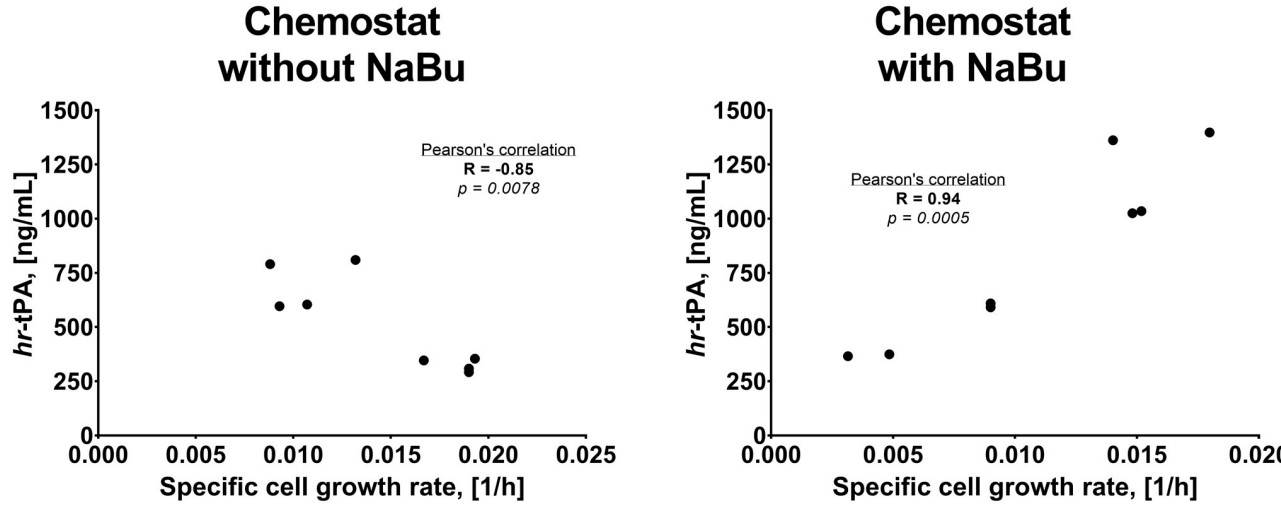

**Fig 3. Relationship between specific cell growth rate and hr-tPA production in CHO cell cultures.** (*right*) Chemostat cultures without sodium butyrate; (*left*) Chemostat cultures with sodium butyrate. Each dot corresponds to one biological replicate. Pearson's correlation was calculated.

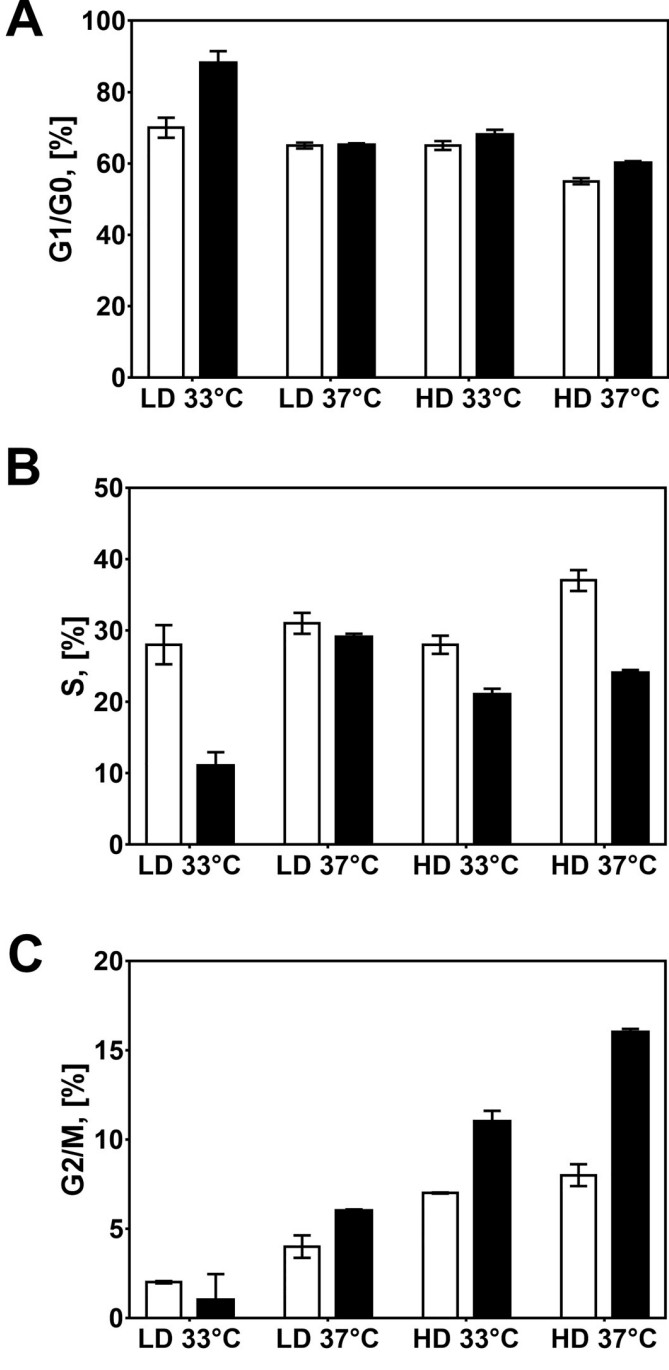

**Fig 4. Cell cycle analysis of CHO cells in continuous culture.** A) Percentage of cells in G1/G0 phase. B) Percentage of cells in S phase. C) Percentage of cells in G2/M phase. Experimental values represent the mean of two biological replicates ± SEM.

cell-specific productivity were strongly correlated with an increase of cells in the G1/G0 phase and a decrease in cells in the S phase (Fig 5A–5D). However, there was no relationship between changes in hr-tPA production and the percentage of cells in the G2/M phase. These data indicated that the increased product titre and $q_P$ were associated with the cell cycle arrest in G1/G0

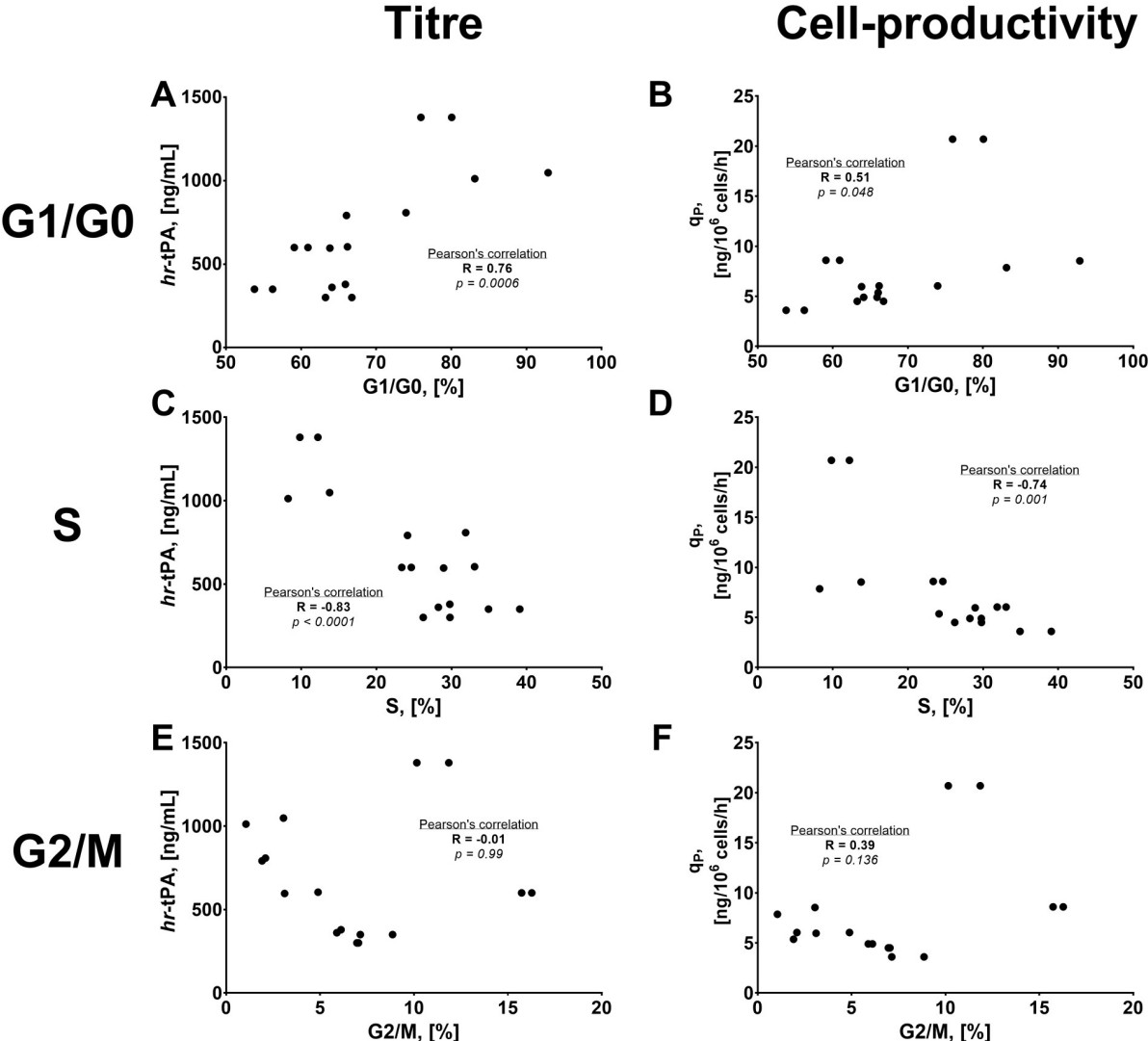

**Fig 5. Relationship between cell cycle distribution and hr-tPA production in CHO cells.** A) Population in G1/G0 phase vs hr-tPA titre. B) Population in G1/G0 phase vs. $q_P$. C) Population in S phase vs hr-tPA titre. D) Population in S phase vs. $q_P$. E) Population in G2/M phase vs hr-tPA titre. F) Population in G2/M phase vs. $q_P$. Each dot corresponds to a biological replicate of a specific culture condition. Pearson's correlations were calculated considering each gene target and their relative titre/$q_P$ corresponding to the control culture conditions (HD-37C-BM).

phase rather than a determined specific cell growth rate. This finding might also explain the differences in the relationship between μ and hr-tPA titres in cultures with and without NaBu.

## Transcriptome analysis of CHO cells in continuous culture at steady state

To better understand the relationship between cell cycle distribution and r-protein production at a molecular level, we performed a transcriptome analysis of CHO cells on samples from our steady-state chemostat cultures using Affymetrix® CHO Gene microarrays. From a total of 26,152 probed genes on the CHO gene array, 5,630 targets were fully identified and annotated with a gene name, functional description, and transcript position. Changes in the gene expression of each target were compared as a ratio of expression between each condition and the control cultures (i.e., HD-37C-BM) (Fig 6). To capture the most reliable data, our analysis

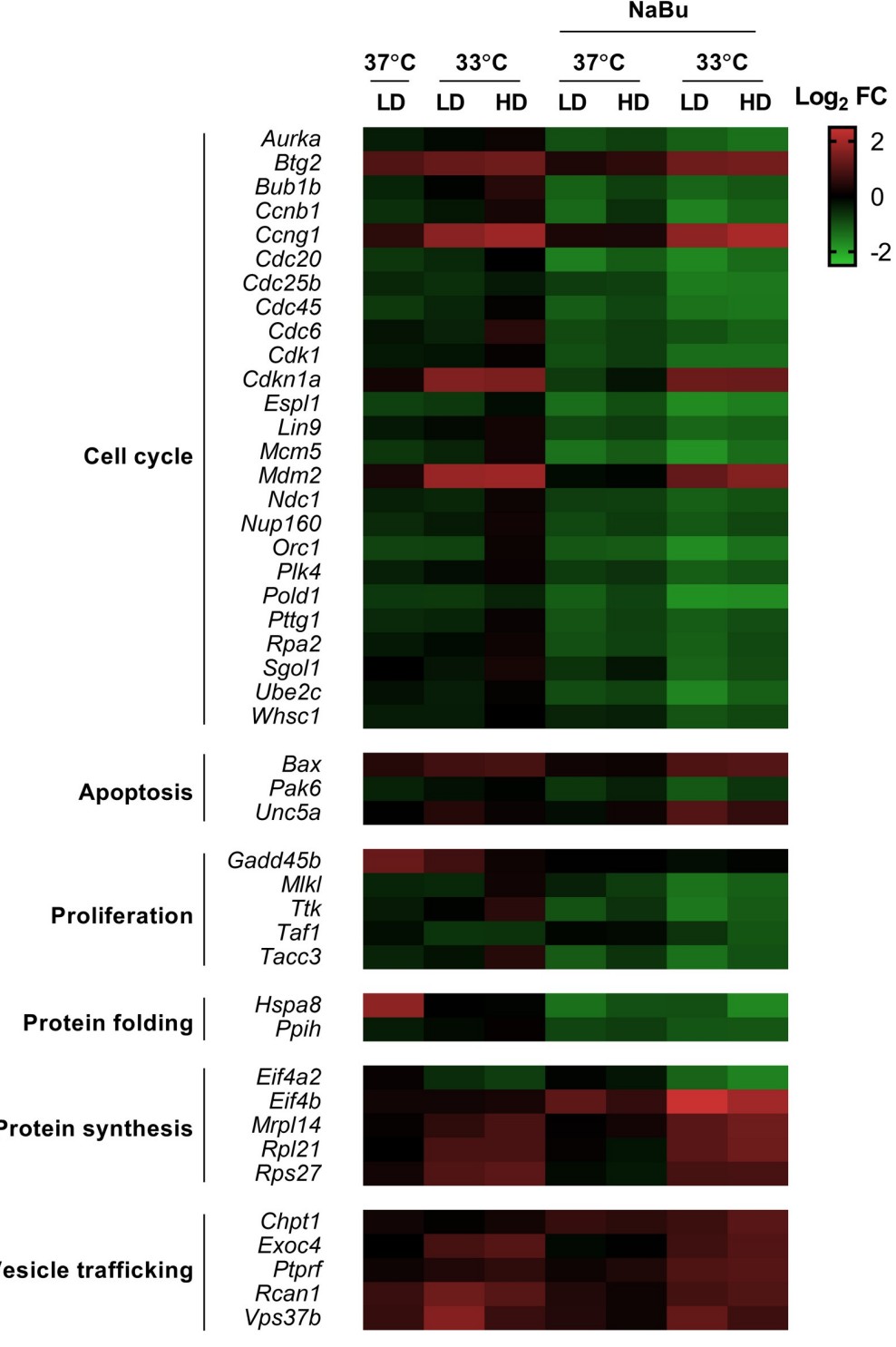

**Fig 6. Heat map of differentially expressed genes.** The mRNA expression levels are shown in Log2FC. Green and red boxes represent down- and upregulated genes, respectively. Genes were grouped into specific biological pathways. Transcriptome analysis is found in S1 File.

included only those identified genes that were upregulated and/or downregulated at least 2-fold (relative to the control), and a *p*-value below 0.01, in the HD-33C-NB cultures (the condition with the highest hr-tPA production). This analysis resulted in a total of 89 genes that were differentially expressed in cells of the HD-33C-NB cultures. Exploration of this high-dimensional dataset using principal component analysis (PCA) showed that low temperature and NaBu supplementation were the conditions that had the major influence on the gene expression profile, with the combination between these two conditions driving the most significant changes (S1 File). Meanwhile, the dilution rate only had minor contributions to the differences observed in the gene expression, and samples did not show major variations in their gene expression profile. Given the minor contribution of the dilution factor on the gene expression profile, our analysis will mainly focus on low temperature and NaBu treated cultures.

A closer examination of the transcriptome changes in response to low temperature and NaBu supplementation indicated that each intervention drastically modified the expression of genes involved in cell growth (i.e., cell cycle, proliferation, and apoptosis) and protein secretion (i.e., folding, translation and vesicle trafficking) (Fig 6). In low temperature cultures, we found a significant increase in *btg2*, *ccng1*, *cdkn1a*, and *mdm2* mRNA expression, a set of genes closely related to the cell cycle progression in the G1 phase and subjected to p53-dependent regulation [30]. The upregulation of *btg2* and *cdkn1a* (also known as p21)—two genes that negatively regulate the transition from G1 to S phase—was consistent with the cell cycle arrest in G1/G0 phase at low temperature. These results also agreed with previous studies of low temperature transcriptome of CHO cells showing activation of p53 gene and associated targets [13]. Low temperature also led to the upregulation of two ribosomal proteins (*rpl21* and *rps27*) and two components of the vesicle secretory pathway (*exoc4* and *vps37b*). The upregulation of vesicle regulatory protein was an event that has been previously observed in low temperature CHO cell cultures [13, 18] and might have contributed to the slight increase in hr-tPA secretion under this condition. Interestingly, we did not find differential expression in transcripts coding for Rbm3 or Cirp, which have been extensively linked to a cold response in different cell lines (including CHO cells) [31, 32]. This discrepancy seemed to be associated with the adaptation process that our CHO cell line underwent previous to the continuous cultures, involving a gradual decrease in culture temperature rather than a sudden temperature downshift.

In the case of NaBu treated cultures, we found a widespread downregulation of genes involved in cell cycle progression and cell division checkpoint. For instance, NaBu supplementation downregulated a series of genes (i.e., *espl1*, *cdc20*, *ccnb1*, *bub1b*, and *pttg1*), regulating the steps between the cell cycle and mitosis. In particular, the decreased mRNA expression of *ccnb1*, which controls the transition from mitosis into G1 phase, might explain the increased number of cells in the G2/M phase compared to the other conditions. Additionally, we observed a decreased expression of genes involved in DNA replication and repair (i.e., *pold1*, *cdc45*, *orc1*, *mcm5*). These findings were associated with the decreased amount of cells in the S phase, where the DNA is replicated and repaired.

Interestingly, we found that most differentially expressed genes responding to low temperature or NaBu supplementation maintained the same pattern when these two approaches were combined. The additive effect of low temperature and NaBu supplementation on the cell cycle arrest in the G1/G0 phase has been previously reported in CHO cells [1]. However, an explanation of the mechanisms underlying this complementary action has not been proposed yet. Our data also indicated that low temperature and NaBu caused cell cycle arrest through different mechanisms. While low temperature led to the upregulation of a series of genes involved in the p53 signaling pathway, NaBu supplementation downregulated targets involved in the

 

**Table 2. List of genes that correlated with a cell cycle arrest in the G1/G0 phase and hr-tPA production in CHO cells.**

| Symbol | Description | Function | G1/G0 arrest | | tPA production | |
|---|---|---|---|---|---|---|
| | | | R | p-value | R | p-value |
| Unc5a | Unc-5 Netrin Receptor A | Inhibition of cell growth | 0.84 | 0.01 | 0.82 | 0.013 |
| Pold1 | DNA Polymerase Delta 1 | DNA replication and repair | -0.74 | 0.035 | -0.79 | 0.019 |
| Cdc25b | Cell Division Cycle 25B | Regulates G1 phase transition | -0.74 | 0.036 | -0.85 | 0.007 |
| Sgol1 | Shugoshin 1 | Protection of centromeric cohesin from cleavage during mitotic prophase. | -0.73 | 0.042 | -0.75 | 0.034 |
| Whsc1 | Nuclear Receptor Binding SET Domain Protein 2 | Histone methyltransferase. | -0.78 | 0.022 | -0.85 | 0.007 |
| | | Involved in promoting proliferation. | | | | |
| Eif4b | Eukaryotic Translation Initiation Factor 4B | Required for the binding of mRNA to ribosomes. | 0.76 | 0.03 | 0.72 | 0.043 |

mitosis and transition into the G1 phase. These differences might explain the additive effects on the cell cycle progression when these two approaches were combined.

We used a statistical analysis to determine correlations between differentially expressed genes and arrest of the cell cycle in the G1/G0 phase and/or hr-tPA titres. The dataset was filtered for correlation with R-values above 0.7 and *p*-values below 0.05 for both G1/G0 arrest vs. targets and hr-tPA titres vs. targets comparisons. This analysis resulted in six genes that were correlated with the cellular phenotypes observed in our cultures. We found that Unc5a and Eif4b mRNA expression were positively linked to the arrest in G1/G0 phase and changes in hr-tPA production (Table 2). Unc5a acts as a tumor suppressor of different cancers through inhibition of cell growth [33, 34]. Eif4b is a key piece for mRNAs translation into proteins and also regulates the translation of proliferative and pro-survival mRNAs [35, 36]. In contrast, we observed four genes (*pold1*, *cdc25b*, *sgol1*, and *whsc1*) with strong negative correlations to arrest of the cell cycle in G1/G0 phase and/or hr-tPA titres. These four genes have relevant functions for the cell cycle progression (particularly the G1 phase) and the initiation of mitosis [37], so their downregulation might contribute to the cell growth phenotypes of our CHO cells in culture.

## Discussion

Low temperature and NaBu supplementation are two of the most used productivity-enhancing approaches in CHO cell cultures. However, our understanding of the molecular mechanisms underlying these improvements remains limited. One of the main reasons for this lack of knowledge lies in the growth-inhibiting effects of low temperature and NaBu on CHO cells, thus opening the question of whether their productivity-enhancing effects result from their impact on cell proliferation or the induction of specific productivity-related genes [17]. Here, we used continuous cultures as a strategy that enables the control of cell-specific growth rate and the examination of the differential effect of low temperature and NaBu supplementation on culture performance and gene expression profile. Our findings indicated that low temperature and NaBu treatment significantly impacted on cell proliferation and hr-tPA production, and these changes in culture performance were associated with the cell cycle progression and its regulation at the transcriptional level.

The reciprocal relationship between cell growth and r-protein production has been extensively described in CHO cell cultures, particularly when subjected to low temperature and NaBu supplementation [1, 10, 38]. Nevertheless, the influence of cell-specific growth rates on productivity phenotypes has been a topic of discussion that might be controversial, as not always growth-inhibiting approaches have led to enhanced r-protein production [25], or

selecting for clones with limited cell growth does not translate in high-producing cell lines (data not published). Our findings indicated that the productivity-enhancing effects of low temperature and/or NaBu resulted from the control of cell cycle progression, particularly the arrest in the G1 phase, rather than the decrease in cell-specific growth rate (Figs 2 to 5). To our knowledge, this is the first study decoupling the effect of a decreased cell-specific growth rate (due to low temperature, NaBu supplementation and/or cultures conditions) on the cell cycle distribution in CHO cells. The relationship between arrested cell cycle in G1/G0 phase and increased r-protein production was also in agreement with other strategies focused on the control of cell cycle progression in CHO cells, both using chemical inhibitors of the cell cycle [39] or genetic engineering approaches [7, 9]. However, it is worth noticing that our cultures with the highest number of cells in the G1/G0 phase did not present the highest hr-tPA production, thus suggesting that other aspects influenced the r-protein production (e.g., protein synthesis, secretory pathway) in low temperature or NaBu-treated cultures.

A key element underlying the pivotal changes in growth/production phenotypes is the change in cell transcriptome, in which genes influencing cell proliferation and cell-specific productivity seem to be regulated in an antagonistic manner. This study showed that low temperature and NaBu supplementation led to changes in the cellular transcriptome, particularly for the cell cycle regulation that correlated with the cellular phenotype observed in our continuous cultures (Fig 6). Our data indicated that the transcriptome associated with the cell cycle and proliferation were different between low temperature and NaBu treated cultures, suggesting that these two approaches controlled the cell cycle through different mechanisms.

Low temperature seemed to arrest the cell cycle in a p53-dependant manner. P53 is a master regulator of the cell cycle and apoptosis, and its activation leads to cell cycle arrest by upregulation of a series of P53 target genes (e.g., *Cdkn1a*, *Bax*, *Mdm2*) [30]. Although our transcriptome analysis did not present any differential expression of p53 in CHO cells at low temperature, we observed activation of several P53-related genes (Fig 6). Additionally, other transcriptome and proteome analyses of CHO cells in low temperature cultures showed a significant upregulation of p53 gene [13, 19]. This discrepancy seemed to be associated with the approach of decreasing culture temperature. While most studies involve a drastic temperature downshift in the middle of the exponential phase, our cells were adapted to low temperature for several culture passages and extended time during the continuous operation until cultures reached a steady state. It seemed the high expression of P53 responded to the drastic change in the culture environment, but its downstream effectors maintained its upregulation in cold-adapted cells. This data also agreed with previous reports in CHO cells overexpressing Cdkn1a, which presented a significant arrest in G1/G0 phase and increased r-protein production [8, 40] (a phenotype similar to the one observed in our cultures).

In contrast, NaBu supplementation led to cell cycle arrest by modulating the expression and activity of cyclin-dependent kinases (CDKs) and inhibitors of the CDKs, which are cell cycle regulatory proteins involved in the G phases transition [41]. NaBu has also been shown to inhibit the activity of histone deacetylases (HDACs), enzymes that are crucial for accurate chromosome segregation during the anaphase, and its inhibition cause disruption of the mitosis and the cell cycle division [42]. Consistent with these studies, we found an extended downregulation of cdks and cell division cycle genes (Fig 6), suggesting that NaBu acted at an effector level rather to an upstream transcriptional regulator level (as the case of low temperature). We also observed an additive effect of low temperature and NaBu on the arrest of the cell cycle in the G1/G0 phase, thus suggesting that their mechanisms controlling cell proliferation can be complementary.

Using correlation analysis, we observed six genes strongly correlated with the cell cycle arrest in G1/G0 and the hr-tPA production (Table 2). These findings suggested that

engineering the expression of these genes might induce cell phenotypes similar to those observed in our low temperature or NaBu treated cultures, without manipulating culture conditions. However, an important caveat to this approach is that altering the gene expression of these targets might limit the capacity to isolate clones, given the antiproliferative effect of these proteins. With an increasing number of synthetic biology tools, it seems possible to develop genetic circuits that enable the switch ON/OFF of these targets as a strategy to engineer the CHO cell machinery capable of controlling the growth/production phases on-demand.

This study has provided novel insights into the regulation of the cell cycle and its relationship to the productive phenotypes of CHO cells. Nonetheless, these results must be interpreted with caution, and potential limitations should be borne in mind. An initial limitation concerns the use of chemostat cultures, a cultivation method that does not represent the reality of current industrially-relevant manufacturing settings, where cells can reach high cell density and product yields. However, continuous culture represents an excellent tool for investigating the isolated effects of environmental parameters on cell physiology and performance. Additionally, findings made in continuous cultures have been shown to be relevant in other cultivation systems [25, 43, 44]. Another important consideration of using continuous culture is the cellular adaptation to the feeding regime. This adaptation often leads to changes in cell density, r-protein production and metabolism during the initial residence times (time when medium is renewed in the vessel) (**S2 Fig in** S2 File). In our experimental design, we consider that steady state was attained when the concentration of viable cells and metabolites are in a 5% deviation range between consecutive points. Our data showed that four residence times enabled the achievement of steady state conditions. Whilst we found small variation in glucose and lactate at HD-33°C conditions (despite the constant cell density and hr-tPA production), these slight deviations did not disrupt the steady state conditions and the biological effects of both low temperature and sodium butyrate observed in this study.

A second potential limitation lies in the use of DNA microarrays. Currently, the advances in DNA/RNA sequencing technology, coupled with the most recent updated genome annotations, enable the precise identification and quantification of transcripts, an aspect that is highly limited in DNA microarrays. However, DNA microarrays have been extensively used for gene expression profiling in CHO cells, and their findings have provided substantial advances in our current understanding of CHO cell physiology. Additionally, given the large number of conditions evaluated in this study, DNA microarrays provided a cost-effective approach for evaluating multiple conditions simultaneously.

In summary, our findings revealed that the productivity-enhancing effects of low temperature and NaBu supplementation were associated with controlling the cell cycle progression. Additionally, this study showed valuable information about the relationship between cell growth and r-protein production in CHO cell cultures and the potential gene markers that can be used for engineering cell lines to control phenotypes during cell line development.

## Supporting information

**S1 File. Additional information 1.**
(XLSX)

**S2 File. Additional information 2.**
(PDF)

## Author Contributions

**Conceptualization:** Mauro Torres, Claudia Altamirano.

**Data curation:** Mauro Torres.

**Formal analysis:** Mauro Torres, Norma A. Valdez-Cruz, Cristian Acevedo.

**Funding acquisition:** Maria Molina Sampayo, Claudia Altamirano.

**Investigation:** Verónica Avello, Mauro Torres.

**Methodology:** Cristian Acevedo.

**Project administration:** Maria Molina Sampayo, Claudia Altamirano.

**Resources:** Claudia Altamirano.

**Software:** Cristian Acevedo.

**Supervision:** Claudia Altamirano.

**Visualization:** Mauro Torres.

**Writing – original draft:** Mauro Torres.

**Writing – review & editing:** Mauricio Vergara, Julio Berrios, Norma A. Valdez-Cruz, Cristian Acevedo, Maria Molina Sampayo, Alan J. Dickson, Claudia Altamirano.

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
