## [Decision Letter · Decision Letter 0]

19 Aug 2022

PONE-D-22-10823Enhanced recombinant protein production in CHO cell continuous cultures under growth-inhibiting conditions is associated with an arrested cell cycle in G1/G0 phase.PLOS ONE

Dear Dr. Altamirano,

Thank you for submitting your manuscript to PLOS ONE. After careful consideration, we feel that it has merit but does not fully meet PLOS ONE’s publication criteria as it currently stands. Therefore, we invite you to submit a revised version of the manuscript that addresses the points raised during the review process.

To accept your manuscript for publication, we ask that you provide time profiles for the continuous cultures (e.g.,  VCD, Glc, Lac, and Titre), indicating when during the culture samples for cell cycle phase and transcriptomic analyses were taken. We ask for this to allow readers to ascertain that the cultures indeed reached steady state conditions.

We look forward to receiving your revised manuscript.

Kind regards,

Ioscani Jimenez del Val, Ph.D.

Academic Editor

PLOS ONE

Journal Requirements:

a) Did participants provide their written or verbal informed consent to participate in this study?

“This work was financially supported by funding (1200962 and 11190488) Fondo Nacional de Desarrollo Cientifico y Tecnologico, (180146) REDES from the National Research and Development Agency of Chile, and (BB/N022041/1) from the Biotechnology and Biological Sciences Research Council of the UK.”

“This work was financially supported by funding (1200962 and 11190488) Fondo Nacional de Desarrollo Cientifico y Tecnologico, (180146) REDES from the National Research and Development Agency of Chile, and (BB/N022041/1) from the Biotechnology and Biological Sciences Research Council of the UK.”

“This work was financially supported by funding (1200962 and 11190488) Fondo Nacional de Desarrollo Cientifico y Tecnologico, (180146) REDES from the National Research and Development Agency of Chile, and (BB/N022041/1) from the Biotechnology and Biological Sciences Research Council of the UK.”

Additional Editor Comments (if provided):

Based on reviewer comments, I recommend that the authors consider including time profiles for the continuous cultures performed under different treatments (mild hypothermia, NaBu, and control). This will allow the readers to confirm that steady states were indeed achieved and will strengthen the conclusions presented by the authors. It may be as simple as adding a supplementary file with the data and referencing it in the manuscript. Once data confirming steady state during continuous cultures is made available, I the manuscript will be ready for publication in PLoS One.

Reviewers' comments:

Reviewer's Responses to Questions

**Comments to the Author**

1. Is the manuscript technically sound, and do the data support the conclusions?

Reviewer #1: Partly

Reviewer #2: Yes

2. Has the statistical analysis been performed appropriately and rigorously? 

Reviewer #1: Yes

Reviewer #2: Yes

3. Have the authors made all data underlying the findings in their manuscript fully available?

Reviewer #1: No

Reviewer #2: Yes

4. Is the manuscript presented in an intelligible fashion and written in standard English?

Reviewer #1: Yes

Reviewer #2: Yes

5. Review Comments to the Author

Reviewer #1: Manuscript Number: PONE-D-22-10823

Manuscript Title: Enhanced recombinant protein production in CHO cell continuous cultures under growth-inhibiting conditions is associated with an arrested cell cycle in G1/G0 phase.

PLOS ONE

This paper focused on the gene expression under continuous CHO cultivation. It seems that this paper contains valuable information. I think that the manuscript should be accepted with major revision.

Chemostat data should be shown in the manuscript.

Authors mentioned that all chemostat cultures were operated at a steady-state. All analytical data were analyzed on the basis of chemostat steady-state. Without this information, it is very difficult to elucidate the obtained data. However, detail time course of cell and limiting-substrate concentrations were not shown. To confirm chemostat steady-state, the detail time course should be shown in the revised manuscript.

Reviewer #2: This is a good paper that examines the mechanism of the effect of low temperature and NaBu, which is considered effective in a production system using CHO cells. It is especially excellent that it has been validated in CHEMOSTAT culture.

6. PLOS authors have the option to publish the peer review history of their article (what does this mean?). If published, this will include your full peer review and any attached files.

Reviewer #1: No

Reviewer #2: No

---

## [Author Response · Author response to Decision Letter 0]

12 Sep 2022

Editor comments: 

“Based on reviewer comments, I recommend that the authors consider including time profiles for the continuous cultures performed under different treatments (mild hypothermia, NaBu, and control). This will allow the readers to confirm that steady states were indeed achieved and will strengthen the conclusions presented by the authors. It may be as simple as adding a supplementary file with the data and referencing it in the manuscript. Once data confirming steady state during continuous cultures is made available, I the manuscript will be ready for publication in PLoS One.”

A: We have included the data regarding our chemostat cultures during the steady state as requested. This data was included in the supplementary file and mentioned in the manuscript to provide the readers an understanding how cells behaved during the continuous cultivation.

Reviewer #1: 

General comments: 

“This paper focused on the gene expression under continuous CHO cultivation. It seems that this paper contains valuable information. I think that the manuscript should be accepted with major revision. Chemostat data should be shown in the manuscript. Authors mentioned that all chemostat cultures were operated at a steady-state. All analytical data were analyzed on the basis of chemostat steady-state. Without this information, it is very difficult to elucidate the obtained data. However, detail time course of cell and limiting-substrate concentrations were not shown. To confirm chemostat steady-state, the detail time course should be shown in the revised manuscript”.

A: Thank you for the time invested in the review of this manuscript and for the positive comments. We have included the data that supported the achievement of a steady state in our chemostat cultures. This data includes profiles of cell growth, hr-tPA production, glucose and lactate throughout the cultures. We included this data in the Supplementary file as S2 Fig, and mentioned in the main manuscript.

Reviewer #2: 

General comments: 

“This is a good paper that examines the mechanism of the effect of low temperature and NaBu, which is considered effective in a production system using CHO cells. It is especially excellent that it has been validated in CHEMOSTAT culture”.

A: We thank the reviewer for the time taken to review this paper and for the positive comments.

---

## [Decision Letter · Decision Letter 1]

12 Oct 2022

PONE-D-22-10823R1Enhanced recombinant protein production in CHO cell continuous cultures under growth-inhibiting conditions is associated with an arrested cell cycle in G1/G0 phase.PLOS ONE

Dear Dr. Altamirano,

Thank you for submitting your manuscript to PLOS ONE. After careful consideration, we feel that it has merit but does not fully meet PLOS ONE’s publication criteria as it currently stands. Therefore, we invite you to submit a revised version of the manuscript that addresses the points raised during the review process.

Minor revisions should be included to briefly discuss deviations from steady state and potential implications on the broader conclusions (i.e., that cell cycle arrest underlies increased productivity) presented in the manuscript.

We look forward to receiving your revised manuscript.

Kind regards,

Ioscani Jimenez del Val, Ph.D.

Academic Editor

PLOS ONE

Journal Requirements:

Additional Editor Comments (if provided):

The authors have successfully addressed all comments made by the reviewers.

However, additional questions have been raised regarding the existence of a true steady state during the continuous cultures.

As suggested by Reviewer 1, minor revisions should be included in the manuscript to briefly discuss deviations from steady state and potential implications on the broader conclusions (i.e., that cell cycle arrest underlies increased productivity) presented in the manuscript.

Whit these minor revisions, the manuscript will be ready for publication in PLoS One and will be of high interest to the journal's readership and the broader CHO community.

Reviewers' comments:

Reviewer's Responses to Questions

**Comments to the Author**

1. If the authors have adequately addressed your comments raised in a previous round of review and you feel that this manuscript is now acceptable for publication, you may indicate that here to bypass the “Comments to the Author” section, enter your conflict of interest statement in the “Confidential to Editor” section, and submit your "Accept" recommendation.

Reviewer #1: (No Response)

2. Is the manuscript technically sound, and do the data support the conclusions?

Reviewer #1: No

3. Has the statistical analysis been performed appropriately and rigorously? 

Reviewer #1: Yes

4. Have the authors made all data underlying the findings in their manuscript fully available?

Reviewer #1: No

5. Is the manuscript presented in an intelligible fashion and written in standard English?

Reviewer #1: Yes

6. Review Comments to the Author

Reviewer #1: This paper focused on the gene expression under continuous CHO cultivation. It seems that this paper contains valuable information. I think that the manuscript should be accepted with minor revision.

Steady-state is not clear

According to time course of chemostat culture as shown in S2 Fig., I think that it is not steady-state.

For example, VCD, lactate concentrations, other concentrations are not constant.

Also, viability data is necessary. Viability (Nt/Nv) should be constant for steady-state.

Limiting-substrate is not clear. According to the difference of limiting substrate, the steady-state should be changed (i.e.,doi: 10.1007/s00449-009-0351-8). Authors should clearly mention it and discuss about it in the revised manuscript.

7. PLOS authors have the option to publish the peer review history of their article (what does this mean?). If published, this will include your full peer review and any attached files.

Reviewer #1: No

---

## [Author Response · Author response to Decision Letter 1]

25 Oct 2022

Editor comments: 

The authors have successfully addressed all comments made by the reviewers. However, additional questions have been raised regarding the existence of a true steady state during the continuous cultures. As suggested by Reviewer 1, minor revisions should be included in the manuscript to briefly discuss deviations from steady state and potential implications on the broader conclusions (i.e., that cell cycle arrest underlies increased productivity) presented in the manuscript. With these minor revisions, the manuscript will be ready for publication in PLoS One and will be of high interest to the journal's readership and the broader CHO community.

A: Thank you for the time taken for reviewing this paper and the positive comments. We addressed the concerns of Reviewer #1 in the section below.

Reviewer #1: 

General comment:

This paper focused on the gene expression under continuous CHO cultivation. It seems that this paper contains valuable information. I think that the manuscript should be accepted with minor revision. Steady-state is not clear. According to time course of chemostat culture as shown in S2 Fig., I think that it is not steady-state. For example, VCD, lactate concentrations, other concentrations are not constant. Also, viability data is necessary. Viability (Nt/Nv) should be constant for steady-state. Limiting-substrate is not clear. According to the difference of limiting substrate, the steady-state should be changed (i.e.,doi: 10.1007/s00449-009-0351-8). Authors should clearly mention it and discuss about it in the revised manuscript.

A: We agree with the reviewer that Fig S2 did not show clearly the steady state, although we believe that this issue was associated with the number of conditions plotted in the graphs that made it difficult to identify the steady state from the adaptation process to the feeding regime. To address this point, we have modified Fig S2 (see below new figure) to show the initiation of the feeding regime, the corresponding adaptation process and the steady state. Cell viability was above 95% for all cultures and showed in FigS1 for the steady state.

We also agree with the reviewer and the editor that the slight variations during the steady state, particularly for glucose and lactate in low temperature at high dilution rate should be discussed. We have addressed this point in the discussion section.

Discussion section (Lines 441-457, Pages 13-14):

“This study has provided novel insights into the regulation of the cell cycle and its relationship to the productive phenotypes of CHO cells. Nonetheless, these results must be interpreted with caution, and potential limitations should be borne in mind. An initial limitation concerns the use of chemostat cultures, a cultivation method that does not represent the reality of current industrially-relevant manufacturing settings, where cells can reach high cell density and product yields. However, continuous culture represents an excellent tool for investigating the isolated effects of environmental parameters on cell physiology and performance. Additionally, findings made in continuous cultures have been shown to be relevant in other cultivation systems [25,43,44]. Another important consideration of using continuous culture is the cellular adaptation to the feeding regime. This adaptation often leads to changes in cell density, r-protein production and metabolism during the initial residence times (time when medium is renewed in the vessel) (Fig S2). In our experimental design, we consider that steady state was attained when the concentration of viable cells and metabolites are in a 5% deviation range between consecutive points. Our data showed that four residence times enabled the achievement of steady state conditions. Whilst we found small variation in glucose and lactate at HD-33°C conditions (despite the constant cell density and hr-tPA production), these slight deviations did not disrupt the steady state conditions and the biological effects of both low temperature and sodium butyrate observed in this study.”

---

## [Editor Report · Decision Letter 2]

1 Nov 2022

Enhanced recombinant protein production in CHO cell continuous cultures under growth-inhibiting conditions is associated with an arrested cell cycle in G1/G0 phase.

PONE-D-22-10823R2

Dear Dr. Altamirano,

We’re pleased to inform you that your manuscript has been judged scientifically suitable for publication and will be formally accepted for publication once it meets all outstanding technical requirements.

Kind regards,

Ioscani Jimenez del Val, Ph.D.

Academic Editor

PLOS ONE

Additional Editor Comments (optional):

I believe the authors have adequately addressed all reviewer comments/suggestions.

I believe the manuscript is now ready for publication in PLoS One.
---

## [Editor Report · Acceptance letter]

4 Nov 2022

PONE-D-22-10823R2 

Enhanced recombinant protein production in CHO cell continuous cultures under growth-inhibiting conditions is associated with an arrested cell cycle in G1/G0 phase. 

Dear Dr. Altamirano:

I'm pleased to inform you that your manuscript has been deemed suitable for publication in PLOS ONE. Congratulations! Your manuscript is now with our production department. 

Kind regards, 

on behalf of

Dr. Ioscani Jimenez del Val 

Academic Editor

PLOS ONE